# Association between physical intimate partner violence and postpartum contraceptive use in the United States–evidence from PRAMS 2016–2021

Rashida-E Ijdi[1,2]☯*, Janine Barden-O'Fallon[1,2]☯

1 Department of Maternal and Child Health, Gillings School of Global Public Health, University of North Carolina at Chapel Hill, Chapel Hill, North Carolina, United States of America, 2 Carolina Population Center, University of North Carolina at Chapel Hill, Chapel Hill, North Carolina, United States of America

☯ These authors contributed equally to this work.
* ijdi@live.unc.edu

**Data Availability Statement:** This study analyzed Pregnancy Risk Assessment and Monitoring Survey (PRAMS) data received from the U.S. Centers for Disease Control and Prevention (CDC).

## Abstract

### Objective

Intimate Partner Violence (IPV) continues to be a major global public health concern, impacting physical and psychological well-being of individuals, including their reproductive and sexual health. The objective of this study is to examine the association between physical intimate partner violence and the utilization of contraception during the postpartum period in the United States.

### Method

This study used data from the CDC's Pregnancy Risk Assessment Monitoring System (PRAMS) survey study phase 8, covering 2016–2021. The sample included 165,204 women reporting physical IPV during pregnancy or 12 months before their last pregnancy and their postpartum contraceptive use. Descriptive, bivariate, and logistic regressions were used to analyze the relationship between IPV and postpartum contraceptive use, adjusting for relevant factors and addressing sampling weights.

### Results

The study found a 3.2% prevalence of physical IPV, with state variances ranging from 2.2% to 5.5%. Among women who experienced physical IPV, 91.0% used contraception, compared to 94.5% of those who did not experience physical IPV. Experiencing physical IPV significantly decreased the likelihood of using any postpartum contraceptive method by 42% (aOR: 0.58; 95% CI: 0.48–0.70) compared to those who did not experience physical IPV during the same period, after adjusting for covariates. Factors that increased the probability of using contraception during the postpartum period included women's higher educational attainment, being married or cohabitating, being employed anytime during pregnancy, and having an unintended last pregnancy.

The data sources used in the analysis can be accessed at the weblink upon request: https://www.cdc.gov/prams/prams-data/researchers.htm.

**Funding:** The author(s) received no specific funding for this work.

**Competing interests:** The authors of this manuscript have declared that no competing interests exist.

**Abbreviations:** aOR, adjusted odds ratio; CI, confidence intervals; CPR, contraceptive prevalence rate; IPV, intimate partner violence; IUD, intrauterine device; PRAMS, pregnancy risk assessment monitoring system; VIF, variance inflation factors; US, United States.

## Conclusion

This study highlights the significant association between physical IPV and reduced use of postpartum contraception in the United States. It calls for the integration of IPV considerations into public health policies and clinical initiatives to improve maternal well-being.

## Introduction

Intimate partner violence (IPV) is a serious though preventable public health problem among women of reproductive age (15–49 years) that can have severe consequences for the trajectory of their lives [1]. According to the Centers for Disease Control and Prevention (CDC), IPV encompasses physical violence, sexual violence, stalking, and psychological aggression by a current or former intimate partner (which may include spouse, boy/girlfriend, dating partner, or sexual partner) [2,3]. Global estimates indicate that about 30% of women who have been in a relationship reported experiencing any physical or sexual IPV at some point in their lives [4]. This percentage varies by region, from 23% in the Western Pacific to 37% in high-income regions of Europe [4]. In the United States (US), about 1 in 4 women has experienced IPV during her lifetime, leading to profound impacts on their physical, mental, and reproductive health [5]. Each year in the U.S., IPV results in over 1,500 deaths [6–8].

Although women of all ages may experience IPV, women of reproductive age are disproportionately affected by physical IPV, leading to various negative health outcomes such as gynecologic disorders, pregnancy complications, unintended pregnancy, and sexually transmitted infections [9]. The lack of control over their reproductive health is identified as a key factor contributing to the elevated risk of unintended pregnancy among abused women [10]. Studies in the United States have shown that 1 in 5 women seeking care in family planning clinics have a history of abuse or have been subjected to pregnancy coercion: with a notable percentage (almost 15%) reporting birth control sabotage [11,12]. Moreover, pregnancy itself is a critical period for the occurrence of IPV. A recent article found that an average of 5.7% of pregnant women were estimated to have experienced any type of IPV during pregnancy in 2016–2018 in the U.S., though the real number could have been much higher than reported [13,14]. The consequences of IPV during pregnancy are severe for both mother and child, including peripartum depression, obstetric complications, preterm birth, low-birth weight infants, and perinatal death [15,16].

Contraceptive utilization is crucial for reducing the likelihood of unintended pregnancy among women experiencing IPV. Research has provided evidence indicating that the primary cause of unintended pregnancy is often attributed to insufficient contraceptive practices [16]. For instance, a survey conducted in the United States revealed that 95% of unintended pregnancies were a result of either not using contraception or using it improperly or inconsistently [17]. Post-partum contraception plays a crucial role in safeguarding women's health by effectively preventing short interpregnancy intervals (<18 months from delivery), reducing the risk of preterm birth, reducing unintended pregnancy, and minimizing associated health complications [18]. However, little is known about factors contributing to contraceptive use among postpartum women who experience IPV around the time of pregnancy. It is anticipated that the effective and consistent use of contraceptives has the potential to decrease the unmet need for postpartum contraception, particularly among populations with a higher prevalence of unintended pregnancies [18].

Attention has been increasingly focused on the impact of physical IPV on the physical and mental well-being of women on a global scale. Specifically, while the contraceptive prevalence rate for women of reproductive age in the U.S. continues to increase, very few studies have aimed to understand the extent to which physical IPV is associated with contraceptive use among postpartum women. Our research expands upon a previous study that investigated the effect of IPV on postpartum contraceptive practices using PRAMS data from 26 states and one city spanning from 2012 to 2015 [19]. There is no recent study to provide information on current changes in physical IPV and postpartum contraception use in the U.S. To further address the knowledge gap, the objective of this study is to use data collected between 2016–2021, adding all participating states and cities to the analysis, to assess the relationship between physical IPV and contraceptive use among postpartum women in the United States. The rationale of this study is to inform policymakers and stakeholders in planning and implementing policies aimed at improving reproductive and maternal health, with a specific focus on enhancing postpartum contraceptive use among women exposed to physical IPV in the United States.

## Methods

This study involved analyzing secondary data from cross-sectional, retrospective surveys conducted by the Pregnancy Risk Assessment Monitoring System (PRAMS). PRAMS is an ongoing city and state-level, population-based surveillance system that focuses on selected maternal behaviors and experiences that occur before, during, and shortly after pregnancy [20]. The PRAMS initiative was carried out in collaboration with state and local health departments by the CDC Division of Reproductive Health.

### PRAMS data

In this study, data from PRAMS phase 8 (2016–2021) were used, which included 46 states, the District of Columbia, New York City, Northern Mariana Islands, and Puerto Rico [21]. PRAMS phase 8 achieved a response rate threshold of 50% overall during the years 2016–2021, with 49 sites participating in 2021, 50 sites in 2020, 2019, 2018, 2017, and 40 sites in 2016 [21]. However, data collection did not occur in four states—California, Idaho, Ohio, and Nevada– during this phase. The live births in the 50 jurisdictions that participated in PRAMS surveillance phase 8 accounted for 81% of all live births in the U.S. [21]. This is the largest dataset representing information on physical IPV and postpartum contraceptive use behavior among women of reproductive age in the United States.

The PRAMS survey employs a stratified sample extracted from birth certificate records of recent live births in participating sites. This sample is used to collect data on experiences and behaviors before, during, and shortly after pregnancy. The infant's mothers were mailed a questionnaire within 2–6 months following delivery, and those who did not respond to repeated mailings were contacted by telephone. Participants consented prior to completing the PRAMS survey per CDC protocols [22]. A comprehensive list of available datasets for Phase 8 (year 2016–2021), categorized by city and state, is provided in S1 Table as an annex alongside this manuscript. The threshold for the minimum overall response rate was set at 55% for 2016–2017, and 50% for 2018–2021 [22]. Detailed information about sampling and survey methodology can be found elsewhere [22].

**Ethics approval.** The PRAMS Phase 8 (2016–2021) data is released to the public and is provided for use by the approval of PRAMS sites and CDC by request [23]. Researchers requested de-identified PRAMS data by submitting a proposal to the CDC with a data-sharing agreement and a standard application form. The study protocol was approved by the CDC and PRAMS participating state health departments by informed written consent to data request

email on December 1, 2022, and the data was shared within four weeks after this approval. Ethical approval of this study was waived by the UNC Chapel Hill Office of Human Research Ethics Institutional Review Board (IRB) (Study# 22–2814) in a written consent on January 23, 2023, because the study was carried out using publicly available data that was anonymized and free of personally identifiable information. Only members of the research team listed on the application had access to the data.

## Sample

Inclusion criteria for the study encompassed participants who provided comprehensive details regarding IPV experienced during pregnancy or within the 12 months prior to pregnancy, as well as their current contraceptive use status. As a result, approximately 20% of individuals (40,876 of 206,080) who otherwise completed the survey had not responded to questions regarding IPV and/or current contraceptive use and were excluded from our analyses. Additionally, individuals with missing responses for all maternal variables analyzed accounted for less than 5% of the population from any strata and were also excluded. Our final analysis was conducted using data obtained from 165,204 women.

## Measurement

The PRAMS questionnaire, conducted annually, consists of two sections—a set of core questions that are asked by all states (such as preconception health, demographics, physical abuse) are included in the first section. The second section comprises standard questions that are selected from a pretested list developed either by CDC or by states on their own [20]. In addition, interested states have the option to develop additional supplements that can be developed to append to the end of the regular PRAMS survey. The complete PRAMS questionnaires can be accessed online and are also provided in the supplemental material [23].

**Outcome variable.** In this study, the outcome variable was the type of postpartum contraceptive method utilized at the time the survey was taken (2–6 months after most recent live birth). These variable measures were derived from core questions of the PRAMS survey. Women participants were asked question "Are you or your husband or partner doing anything now to keep from getting pregnant?", and if so, asked the following question "What kind of birth control are you or your husband or partner using now to keep from getting pregnant?". This question was used as a basis to determine the outcome variables in this study. Methods of contraception included in the survey were birth control pills, condoms, shots or injections (Depo-Provera), implants (Nexplanon or Implanon), contraceptive patch (OrthoEvra) or vaginal ring (NuvaRing), IUD (including Mirena, ParaGard, Liletta, or Skyla), tubes tied or blocked (female sterilization or Essure, or tubectomy), vasectomy (male sterilization), natural family planning (including rhythm method), withdrawal (pulling out), abstinence, or other methods. The hierarchy of contraceptive effectiveness in descending order was: tubectomy, vasectomy, IUD, implant, shots, or injections (Depo-Provera) every 3 months, contraceptive patch or vaginal ring, birth control pills, condom, abstinence, rhythm, withdrawal, and "other" [24–26].

**Exposure variable.** The primary exposure variable of interest is the experience of physical IPV (yes/no), measured by asking the following questions: "During the 12 months before you got pregnant you're your new baby, did your husband or partner push, hit, slap, kick, choke, or physically hurt you in any other way?" and "During your most recent pregnancy, did your husband or partner push, hit, slap, kick, choke, or physically hurt you in any other way?". Women were classified as having experienced physical IPV around the time of pregnancy if they answered 'yes' to either question. This is defined as the response to being "pushed, hit,

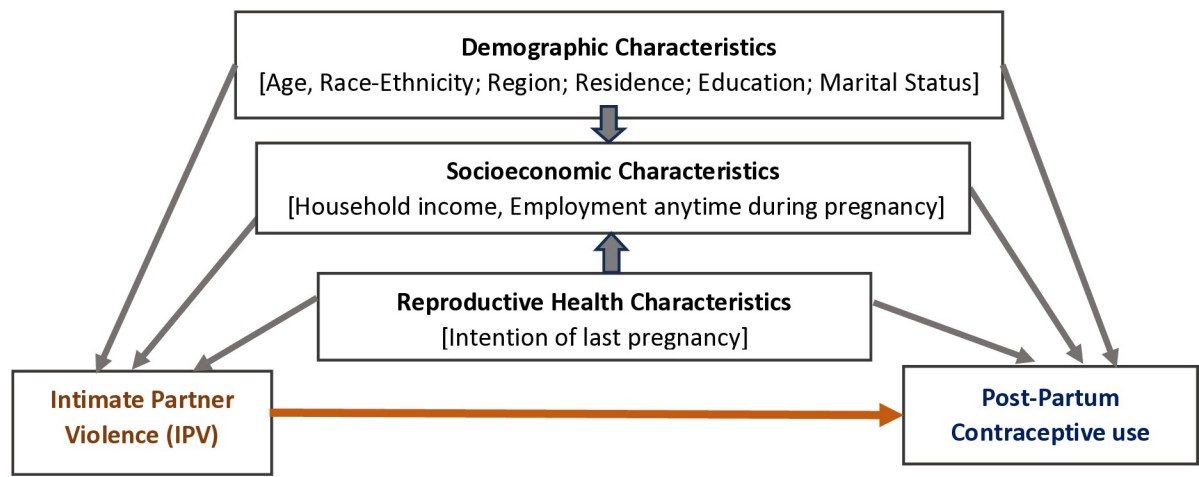

**Fig 1. Direct Acyclic Graph (DAG) of association between IPV and contraceptive use.**

slapped, kicked, choked, or physically hurt" by a current or former husband or partner during the 12 months before pregnancy, during pregnancy, and since the new baby was born.

**Covariates.** Demographic and socioeconomic factors for the analysis were selected based on prior literature, preliminary analysis, and the Direct Acyclic Graph (DAG) diagram [27] (Fig 1). These variables included maternal age (categorized as < 24, 25–29, 30–34, and ≥ 35 years), race/ethnicity (categorized as 'Black non-Hispanic', 'White non-Hispanic', all other races (e.g., Asian, American-Indian, Chinese, Japanese, Philipino, mixed races, other non-White) as 'Other, non-Hispanic', and 'Hispanic'), marital status (married/cohabitated or other), place of residence (urban/rural), educational attainment (< 12 grade, high school graduate, college/beyond), and employment status at any time during last pregnancy (yes/no). The 'state of residence' variable was categorized into four geographic regions based on US Census divisions: Northeast, Midwest, South, and West [28], with the aim to reduce the potential confounding effects of individual state-level differences that many not be pertinent to the scope of this study. Connecticut, Rhode Island, Massachusetts, New Hampshire, Vermont, Maine, New York, Pennsylvania, and New Jersey are in Northeast region; North Dakota, South Dakota, Nebraska, Kansas, Minnesota, Iowa, Missouri, Wisconsin, Illinois, Michigan, Indiana, and Ohio are included in the Midwest region; Oklahoma, Texas, Arkansas, Louisiana, Mississippi, Alabama, Tennessee, Kentucky, Florida, Georgia, YC, North Carolina, Virginia, West Virginia, Maryland, Delaware, District of Columbia, and Puerto Rico are in the South geographic region; and Alaska, Washington, Oregon, Idaho, Montana, Wyoming, California, Nevada, Utah, Arizona, New Mexico, Colorado, and Hawaii are in the West geographic region [28]. Household income in the 12 months before last live childbirth was categorized as 'less than or equal to the average median household income' or 'above average median household income' created from the poverty scale based on the HHS Poverty Guidelines for years 2016–2021 [29]. On average, the median household income over 2016–2021 was approximately $69,878 [29]. Additionally, a reproductive health variable was included, with the intentionality of the last pregnancy, categorized as yes for intended pregnancy, and no for unintended pregnancy. An 'intended pregnancy' was characterized by the respondent wanted to be pregnant at that time or sooner and an 'unintended pregnancy' was described as a pregnancy that was wanted at a later time, not wanted at that moment or at any point in the future, or 'uncertain/not sure'.

All variables were derived from the linked birth certificate. Only annual household income was obtained from the PRAMS survey questionnaire. All measures are current status (i.e., at the time of the survey), unless stated otherwise.

## Statistical analyses

Bivariate analyses were conducted to observe the distribution of each covariate by the explanatory variable. Among participants in the overall sample, the prevalence of contraceptive use (including method-mix) and IPV were estimated using Rao-Scott $\chi^2$ tests. Separate measurements using Rao-Scott $\chi^2$ tests were conducted to assess the association between demographic and socioeconomic characteristics with IPV and the use of contraceptive methods. A model of the relationship between IPV and the use of contraception was developed using multiple logistic regression.

In specifying our model, we have incorporated a directed acyclic graph (DAG) [27] presented in Fig 1, to recognize potential confounders and determine a minimally sufficient adjustment set [30,31].

Given the large number of covariates, multicollinearity was assessed using Variance Inflation Factors (VIF). All covariates were acceptable with VIFs below 5, [32] including our calculated VIF of 2.36. Given that there were no significant correlations between variables, all covariates were included in the final model.

All analyses were conducted using statistical software package STATA, Version 18.0 [33] to account for the complex sampling survey design of PRAMS. Data were weighted for sample design, nonresponse, and non-coverage, and represented individuals residing in the respective states who had live birth during the survey period [21]. The results in the tables were presented with weighted percentages and unweighted numbers.

## Results

Experience of IPV, as well as demographic and socioeconomic characteristics of women who gave birth in the United States between 2016 and 2021 are provided in **Table 1**. According to the data collected from 46 states, the District of Columbia, New York City, and Puerto Rico, the overall prevalence of reporting physical IPV during and/or 12 months before pregnancy was 3.2%. The majority of women in the sample were aged between 20 and 39, accounting for 93% of the participants. In terms of racial and ethnic identification, 14% identified as Black non-Hispanic, 58% identified as White non-Hispanic, 19% identified as Hispanic, and 9% identified as belonging to other racial/ethnic groups. The 'Other' race was categorized as Other Asian (4.3%), American Indian (3.6%), Chinese (1.2%), Japanese (0.2%), Philipino (0.8%), Hawaiian (0.1%), Ak Native (0.8%), mixed race (5.7%), and other non-white (5.5%). Women from the South constituted 41% of the sample, while 84% of the participants resided in urban areas. Approximately 64% of the women had attended college or beyond and had a household income of less than or equal to the 2021 median in 61% of cases. More than one-third of women were unmarried/not cohabiting at the time of the survey (38%). Slightly more than half of the pregnancies (57%) were intended. Women's employment status at any time during last pregnancy was the only variable included in the DAG diagram that did not show a statistically significant difference by IPV status.

**Fig 2** illustrates the variation in the prevalence of physical IPV, ranging from 2.2% (in Connecticut, Georgia, New Jersey, and Washington) to 5.5% (in Arkansas). The Midwest region had the highest reported prevalence of physical IPV at 3.5%, followed by the South at 3.3% and the Northeast at 3.0%. Conversely, West exhibited the lowest prevalence at 2.9%. The national prevalence of physical IPV was 3.2%.

**Table 1. Characteristics of women with a recent live birth, stratified by self-reported exposure to physical IPV during pregnancy or 12 months before last pregnancy in the United States, PRAMS 2016–2021\*.**

| Maternal Characteristics | Exposure to physical IPV | | Total | p-value |
|---|---|---|---|---|
| | Women did not report physical IPV | Women reported physical IPV present | | |
| | % (n) | % (n) | % (N) | |
| **Total** | **96.8 (159,058)** | **3.2 (6,146)** | **100.0 (165,204)** | |
| **Demographic characteristics** | | | | |
| **Age** | | | | |
| Younger than 20 | 3.8 (6,344) | 8.2 (474) | 4.0 (6,818) | <0.001 |
| 20–29 | 47.1 (74,420) | 63.1 (3,740) | 47.6 (78,160) | |
| 30–39 | 45.9 (73,053) | 27.0 (1,813) | 45.3 (74,866) | |
| 40 or older | 3.1 (5,235) | 1.7 (119) | 3.1 (5,354) | |
| **Race-Ethnicity** | | | | |
| Black, Non-Hispanic | 13.9 (26,614) | 21.9 (1,473) | 14.1 (28,987) | <0.001 |
| White, Non-Hispanic | 57.9 (73,882) | 49.7 (2,131) | 57.6 (76,013) | |
| Other, Non-Hispanic | 8.6 (24,005) | 8.4 (1,237) | 8.6 (25,242) | |
| Hispanic | 18.7 (29,989) | 19.0 (1,166) | 18.7 (31,155) | |
| **Region** | | | | |
| Northeast Region | 20.4 (34,887) | 18.8 (1,047) | 20.4 (35,934) | <0.001 |
| Midwest Region | 25.0 (39,813) | 27.1 (1,872) | 25.0 (41,685) | |
| South Region | 40.6 (48,155) | 41.3 (1,869) | 40.6 (50,024) | |
| West Region | 14.0 (36,203) | 12.8 (1,358) | 14.0 (37,561) | |
| **Residence** | | | | |
| Urban | 84.0 (123,580) | 78.1 (4,385) | 83.8 (127,965) | <0.001 |
| Rural | 15.4 (33,091) | 21.4 (1,702) | 15.6 (34,793) | |
| **Educational attainment** | | | | |
| Less than 12[th] grade | 10.8 (17,894) | 17.2 (1,065) | 11.1 (18,959) | <0.001 |
| High school graduate | 24.0 (37,215) | 36.1 (2,179) | 24.4 (39,394) | |
| College or beyond | 64.3 (102,573) | 45.6 (2,840) | 63.7 (105,413) | |
| **Marital status** | | | | |
| Married/cohabiting | 63.3 (97,895) | 23.8 (1,436) | 62.1 (99,331) | <0.001 |
| Other | 36.7 (61,049) | 76.0 (4,698) | 37.9 (65,747) | |
| **Socioeconomic characteristics** | | | | |
| [+]**Household Income, year before delivery** | | | | |
| Less than or equal to median income level[a] | 59.7 (95,092) | 86.1 (5,054) | 60.5 (100,146) | <0.001 |
| Above median income level | 33.2 (52,968) | 7.2 (683) | 32.3 (53,651) | |
| **Employment any time during last pregnancy** | | | | |
| Yes | 44.6 (72,362) | 41.6 (2,621) | 44.5 (74,983) | 0.278 |
| No | 26.1 (39,688) | 25.5 (1,490) | 26.1 (41,178) | |
| **Reproductive health characteristics** | | | | |
| **Intentionality of last pregnancy** | | | | |
| Unintended — Wanted later | 19.2 (29,860) | 26.9 (1,586) | 19.4 (31,446) | <0.001 |
| Unintended — Did not want then or any time | 6.3 (10,249) | 16.7 (997) | 6.7 (11,246) | |
| Unintended — Was not sure | 14.9 (25,293) | 25.1 (1,692) | 15.3 (26,985) | |
| Intended | 58.2 (91,307) | 29.4 (1,744) | 57.3 (93,051) | |

Note. \*Includes 46 US states, New York City, District of Columbia, and Puerto Rico.

[a]Average median income level = $69,878; [+]Yearly total household income, during 12 months before last childbirth.

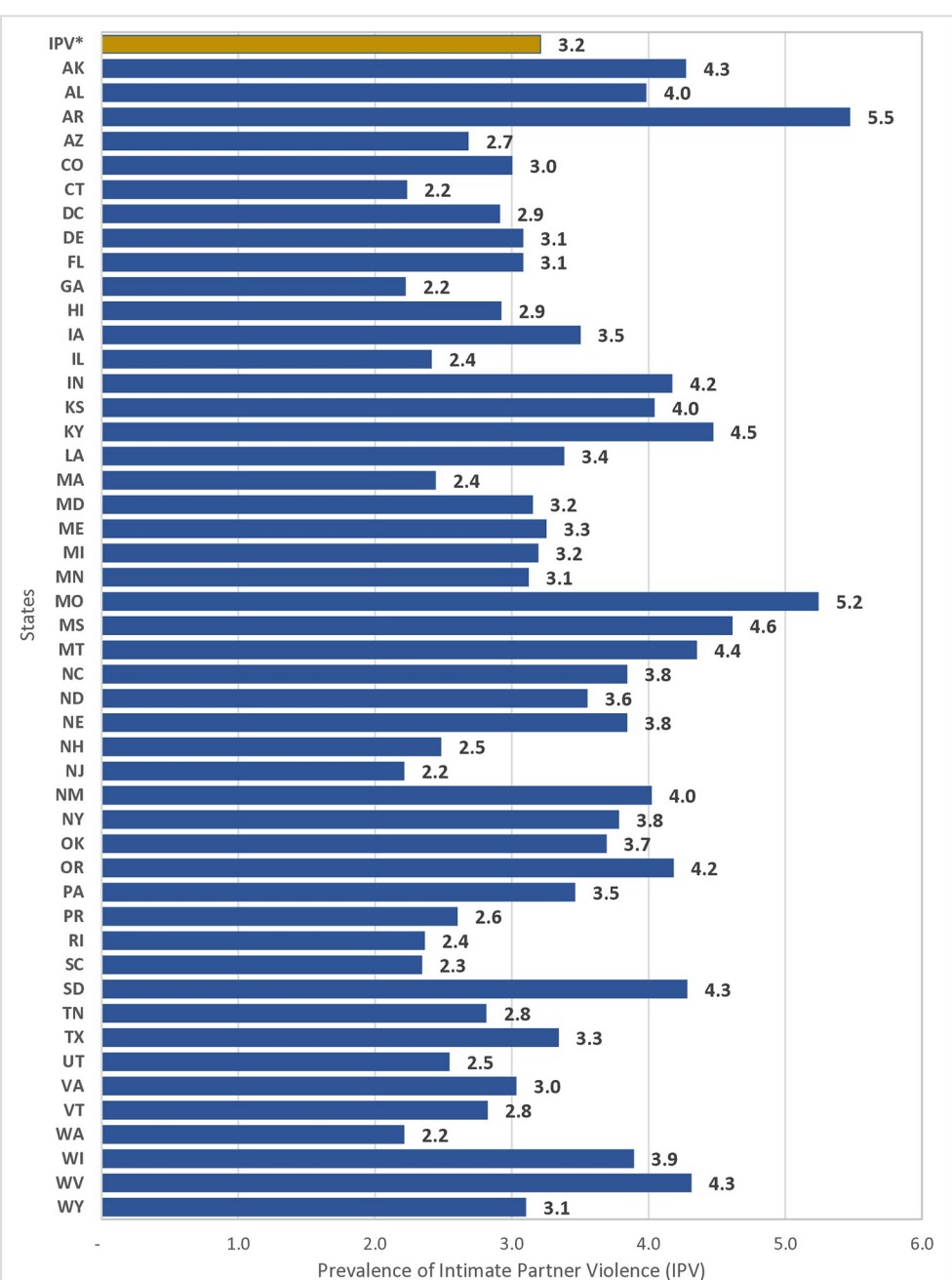

Note: *National Prevalence of Intimate Partner Violence (IPV). Data from 46 states, the District of Columbia, New York City, the Northern Mariana Islands, and Puerto Rico.

**Fig 2. Prevalence of IPV among women with a recent live birth in the United States, PRAMS, 2016–2021.**

**Table 2** illustrates the bivariate analysis of contraceptive use during the postpartum period among women who have and have not experienced physical IPV before and during their last pregnancy. The overall utilization of any method of contraception during the postpartum period was 94%, with a slightly higher rate at 95% among women who had not experienced physical IPV, and a somewhat lower rate at 91% among women who had experienced physical IPV during pregnancy or 12 months before last pregnancy. The most commonly used methods

**Table 2. Postpartum contraceptive use by experience of physical IPV during pregnancy or 12 months before pregnancy in the United States, PRAMS 2016–2021.**

| Contraceptive Method | Exposure to physical IPV | | Total % (N) | p-value |
|---|---|---|---|---|
| | IPV absent % (n) | IPV present % (n) | | |
| | n = 159,058 | n = 6,146 | N = 165,204 | |
| Current contraceptive use during Post Partum period* | 94.5 (149,346) | 91.0 (5,498) | 94.4 (154,844) | <0.001 |
| **Method-mix during postpartum period** | | | | |
| Tubectomy | 8.4 (13,992) | 9.8 (581) | 8.5 (14,573) | <0.001 |
| Vasectomy | 1.8 (2,869) | 0.5 (43) | 1.8 (2,912) | |
| IUD, coil, loop | 13.9 (23,165) | 11.8 (717) | 13.8 (23,882) | |
| Implant | 4.7 (8,856) | 6.6 (475) | 4.7 (9,331) | |
| Shots or injections (Depo-Provera) (every 3 months) | 5.3 (9,383) | 8.7 (526) | 5.4 (9,909) | |
| Contraceptive patch or vaginal ring | 1.3 (2,149) | 1.4 (108) | 1.4 (2,257) | |
| Birth control pills | 14.4 (21,141) | 7.7 (446) | 14.1 (21,587) | |
| Condom | 18.8 (28,144) | 9.0 (511) | 18.5 (28,655) | |
| Abstinence | 10.7 (18,288) | 24.0 (1,554) | 11.1 (19,842) | |
| Rhythm | 2.4 (3,595) | 0.8 (42) | 2.3 (3,637) | |
| Withdrawal | 16.0 (23,784) | 15.2 (865) | 15.9 (24,649) | |
| Other | 2.3 (3,692) | 4.5 (278) | 2.4 (3,970) | |

Note

*p-value = <0.001; Missing values are total 0.53 (882)

were condoms (19%) and birth control pills (14%); while withdrawal (16%) and abstinence (11%) were two commonly used methods. A notable difference between women exposed to IPV and women who were not was observed in the use of birth control pills (14% vs. 8%, respectively) and condoms (19% vs. 9%, respectively), with statistically significant at 5% level ($p<0.001$). For the long-acting-reversible and permanent methods (LARC-PM), IUD and vasectomy use were higher among IVP non-exposed group, whereas implants, shots, and tubectomy use were more common among the IPV-exposed group compared to their counterparts.

Results of the logistic regression models are presented in **Table 3**, which presents both the unadjusted (also known as crude models), and the adjusted model that considered the effect of confounders from different categories, such as, demographic, socioeconomic, and reproductive health factors. In the multivariable regression models, we have selected the category with the lowest prevalence of physical IPV as the reference group for each variable. From the unadjusted models, we found that exposure to physical IPV during pregnancy or 12 months before last pregnancy, mother's age, race-ethnicity, region, urban/rural residence, marital status, employment, and intention of last pregnancy were statistically significant predictors of postpartum contraceptive use without adjusting for other covariates. Interestingly, two important demographic and socioeconomic factors—educational attainment and household income for the year before last childbirth, were not statistically significant in the crude models.

After adjusting for demographic, socioeconomic, and reproductive health variables, our analyses revealed that women who experienced physical IPV during or 12 months before their last pregnancy had 42% lower odds of using any forms of contraceptive methods (aOR: 0.58; 95% CI: 0.48–0.70) compared to women who did not experience physical IPV during the same period. Additionally, we observed that the odds of using postpartum contraception were significantly higher for women aged 20 years or younger compared to women aged 40 or older

**Table 3. Association of IPV with postpartum contraceptive use, unadjusted and adjusted Logistic Regression Models [unweighted N = 165,204], PRAMS 2016–2021.**

| Women's characteristics | Postpartum contraceptive use | | | |
| --- | --- | --- | --- | --- |
| | Unadjusted Model | | Adjusted Model | |
| | OR (95% CI) | *p-value* | OR (95% CI) | *p-value* |
| **Primary variables of interest** | | | | |
| **Physical IPV during pregnancy or 12 months before last pregnancy** | | | | |
| IPV absent | 1.00 | | 1.00 | |
| IPV present | 0.59 (0.50–0.68) | <0.001 | 0.58 (0.48–0.70) | <0.001 |
| **Socio-demographic characteristics** | | | | |
| **Age** | | | | |
| 20 years or younger | 2.43 (1.88–3.14) | <0.001 | 3.13 (2.22–4.41) | <0.001 |
| 20–29 | 1.82 (1.55–2.12) | <0.001 | 2.11 (1.76–2.53) | <0.001 |
| 30–39 | 1.52 (1.30–1.77) | <0.001 | 1.77 (1.49–2.11) | <0.001 |
| 40 or older | 1.00 | | 1.00 | |
| **Race-Ethnicity** | | | | |
| Hispanic | 1.00 | | 1.00 | |
| Black, Non-Hispanic | 0.90 (0.80–0.99) | <0.001 | 0.71 (0.60–0.83) | <0.001 |
| White, Non-Hispanic | 0.90 (0.80–0.90) | 0.041 | 0.87 (0.75–1.01) | 0.067 |
| Other, Non-Hispanic | 0.49 (0.43–055) | <0.001 | 0.45 (0.37–0.52) | <0.001 |
| **Region** | | | | |
| West Region | 1.00 | | 1.00 | |
| Northeast Region | 0.64 (0.57–0.70) | <0.001 | 0.63 (0.55–0.70) | <0.001 |
| Midwest Region | 0.81 (0.73–0.88) | <0.001 | 0.83 (0.73–0.92) | 0.001 |
| South Region | 0.79 (0.71–0.86) | <0.001 | 0.78 (0.69–0.87) | <0.001 |
| **Residence** | | | | |
| Rural | 1.00 | | 1.00 | |
| Urban | 0.89 (0.80–0.97) | 0.017 | 0.96 (0.84–1.08) | 0.498 |
| **Educational attainment** | | | | |
| Less than 12th grade | 1.00 | | 1.00 | |
| High school graduate | 1.05 (0.92–1.19) | 0.465 | 1.17 (0.98–1.39) | 0.072 |
| College or beyond | 1.08 (0.96–1.20) | 0.205 | 1.50 (1.26–1.77) | <0.001 |
| **Marital status** | | | | |
| Married/cohabiting | 1.00 | | 1.00 | |
| Other | 0.92 (0.85–0.98) | 0.017 | 0.76 (0.68–0.84) | <0.001 |
| **Socioeconomic characteristics** | | | | |
| **Household Income, year before last childbirth** | | | | |
| Less than or equal to the average median income level | 0.95 (0.88–1.02) | 0.172 | 0.94 (0.85–1.04) | 0.257 |
| Above average median income level | 1.00 | | 1.00 | |
| **Employment Status any time during last pregnancy** | | | | |
| No | 1.00 | | 1.00 | |
| Yes | 1.18 (1.09–1.28) | <0.001 | 1.15 (1.06–1.25) | 0.001 |
| **Reproductive health** | | | | |
| **Intentionality of last pregnancy** | | | | |
| Intended Pregnancy | 1.00 | | 1.00 | |
| Unintended Pregnancy | 1.09 (1.01–1.17) | 0.015 | 1.20 (1.09–1.32) | <0.001 |

Note: IPV, intimate partner violence; [+]Yearly total household income, during 12 months before last childbirth.

Data are adjusted odds ratio (95% confidence interval) unless otherwise specified.

(aOR: 3.13; 95% CI: 2.22–4.41), indicating a strong positive association between age and post-partum contraceptive use.

Ethnicity also played a role, as Black non-Hispanic women had 29% lower odds of using post-partum contraception (aOR: 0.71; 95% CI: 0.60–0.83) compared to Hispanic women, while other non-Hispanic women had 55% lower odds (aOR: 0.45; 95% CI: 0.37–0.52) as compared to the same. Geographically, women living in the Northeast, Midwest, and South regions had lower odds of contraceptive use compared to those in the West. Education level was a strong positive predictor of contraceptive use, with women who pursued college education or beyond being 50% more likely (aOR: 1.50; 95% CI: 1.26–1.77) to use contraception, compared to those with educational attainment was below 12th-grade. Marital status also had an impact, as unmarried or cohabiting women were 24% less likely (aOR: 0.76; 95% CI: 0.68–0.84) to use contraception compared to married or cohabiting women. Employment during pregnancy also increased the likelihood of contraceptive use by 15% (aOR: 1.15; 95% CI: 1.06–1.25). Women whose last pregnancy was unintended had 20% higher odds of using contraceptive methods during their postpartum period compared to those with an intended pregnancy (aOR: 1.20; 95% CI: 1.09–1.32). Household income and place of residence status did not affect postpartum contraception use

## Discussion

Our study findings suggest that physical IPV during pregnancy or 12 months before pregnancy can have an impact on the use of postpartum contraception. Women who experienced physical IPV during that period were significantly less likely to use postpartum contraception compared to those who did not report physical IPV during the same period; even after controlling for factors associated with post-partum contraceptive use. This finding aligns with existing evidence that physical IPV is associated with adverse sexual and reproductive health outcomes, such as unintended pregnancy, sexually transmitted diseases, or other reproductive health concerns among sexually active post-partum women in the United States [34]. Our research builds on this evidence by demonstrating that physical IPV is associated with nonuse of contraception in the postpartum period, further highlighting the public health importance of addressing IPV in maternal health interventions. Similar results have been observed in earlier studies, such as an analysis of PRAMS data from 2004 to 2008 that found that women who reported physical IPV were significantly less likely to use postpartum contraception than those who did not (cOR, 0.66; 95% CI, 0.61–0.71) [34]; research from 2012 to 2015 also had similar findings [34,35]. Our findings suggest that the likelihood of postpartum contraceptive use among women who have experienced physical IPV has since decreased. Additionally, our analyses demonstrated that the prevalence of IPV was higher among postpartum women who were predominantly younger, of lower socioeconomic status, and in non-marital or cohabiting relationships.

The contraceptive method-mix varied significantly between women who reported physical IPV and those who did not. Our analyses indicated that women who experienced IPV had significantly lower usage rates for partner-dependent contraceptive methods such as condoms, natural family planning (including rhythm), and vasectomy, compared to their counterparts. The reduced usage of these methods could be attributed to partner negotiation, potentially resulting in higher rates of nonuse, as suggested in the CDC's NISVS report [35]. Furthermore, women who have experienced IPV may experience additional challenges to accessing and using effective contraception that is not subject to partner interference and negotiation [36]. However, almost one-quarter of women who had experienced physical IPV during pregnancy or 12 months before pregnancy were abstinent at the time of the PRAMS survey. This disparity

in circumstance may underscore the complex interplay between intimate partner violence (IPV), the dynamics of power within relationships, and the decision-making processes regarding reproduction, thereby highlighting the necessity for interventions that consider these broader contexts [37,38].

The higher rate of nonuse of postpartum contraception among women who have experienced physical IPV during pregnancy or 12 months before last pregnancy, compared to those who had not, indicates an unmet need for family planning resources before, during and shortly after pregnancy. Healthcare providers can play a crucial role in addressing this issue by following physical IPV screening guidelines during pregnancy [37], and providing family planning, education, and resources to all patients, regardless of disclosure [39]. Screening by itself might not fully address the impact of physical IPV on postpartum contraceptive use. However, proactive violence prevention could potentially make a significant difference on this reproductive health outcome. Patient-centered contraceptive counselling by healthcare providers during pregnancy and assistance in accessing desired methods after delivery [40], including contraceptive methods that are less detectable by partners, like implants, IUDs, and injectables, may be particularly beneficial for women experiencing physical IPV [37]. Moreover, the prevalence of physical IPV varies significantly across different states in the USA. Some states have more robust support systems and legal protections for IPV survivors, which can influence reporting rates and overall prevalence of IPV [41].

Our research highlights the association between exposure to physical IPV during pregnancy or 12 months before pregnancy and its influence on postpartum contraceptive use. This underscores the importance of preventing IPV to support women's reproductive autonomy. Our findings also suggest that reducing IPV exposure around the time of pregnancy could empower women to make autonomous contraceptive decisions. Comprehensive IPV interventions, aligned with recommendations from the American College of Obstetricians and Gynecologists (ACOG) [9,40], USPSTF [42], and CDC [3], may play a crucial role in promoting women's overall health and well-being. Ensuring pregnant women's access to comprehensive family planning services can serve as a critical intervention for those experiencing IPV, by supporting their autonomy, and promoting their overall health and well-being.

## Strength and limitations

There are several strengths and limitations to this study. A key advantage of the study is the use of data collected from the largest national probability sample of women who have recently delivered a live infant in the U.S. [43]. Since PRAMS uses a standardized data collection methodology developed by the CDC, the data is reliable. Each state that carried out PRAMS followed the same methodology, while still having the flexibility to adapt the survey tool to meet the state's requirements.

Despite its strengths, the study has several significant limitations that must be acknowledged. Population-based studies face challenges in accurately assessing the extent of physical IPV, as the questions in the PRAMS survey may not be behaviorally specific and could lead to underestimations. Additionally, respondents in the postpartum study may inaccurately recall instances of abuse from years prior. However, only mothers with live births were surveyed by PRAMS. No information was available about mothers who had a miscarriage, stillbirth, or other nonviable pregnancy outcomes. Excluding mothers with non-live births limits the generalizability of the study findings using PRAMS data. Again, self-reported data from new mothers may introduce biases, as participants may not accurately remember events or may be hesitant to disclose information. We decided to focus on physical IPV in this study, because data on psychological and emotional abuse were not included for every state or city in the

survey. Furthermore, excluding a significant number of women due to non-responses could have biased the results in unknown directions. Furthermore, not including information on the timing of contraception initiation complicates the interpretation of results. Moreover, the survey did not provide information on when women initiated contraception, only what method they were using at the time of the survey. Our analysis is based on individuals listed as the birthing parent on the birth certificates and may not fully encompass all gender identities. Most importantly, the lactational amenorrhea method (LAM) was not included in the listed contraceptive methods in the PRAMS questionnaire. We know that LAM is a significantly effective form of postpartum contraception [34,36,37] that provides substantial protection against pregnancy, particularly in the first six months of postpartum when exclusively breast-feeding. The inclusion of LAM could have indeed enriched our understanding of postpartum contraceptive practices among women, especially considering its non-invasive nature and accessibility.

## Conclusion

The study findings reveal the impact of physical IPV on postpartum contraceptive use, highlighting the need for IPV screening during pregnancy including the provision of person-centered post-partum family planning education and counseling, which is vital for ensuring women's reproductive autonomy and well-being. These findings also call for integrating IPV prevention into maternal health strategies and enhancing support for affected women, building on existing policies. Further research should continue to explore the long-term effects of IPV on maternal health to inform clinical and policy initiatives further.

## Supporting information

**S1 Table. PRAMS, Phase 8 (2016–2021) participating states according to data collection years.**
(DOCX)

## Acknowledgments

This article was made possible by thousands of women in the United States who shared their personal experiences by mail or telephone conversations with the research teams throughout the country who carried out the surveys, provided access to data, and prepared the dataset according to the study requirements. We would also like to thank the CDC PRAMS Team, Applied Sciences Branch, Division of Reproductive Health. PRAMS working group representatives listed by State: Alabama—Danita Crear, DrPH; Alaska—Kathy Perham-Hester, MS, MPH; Arkansas—Mary McGehee, PhD; Colorado—Alyson Shupe, PhD; Connecticut—Jennifer Morin, MPH; Delaware—George Yocher, MS; District of Columbia–Pamela Oandasan; Florida—Jerri Foreman, MPH; Georgia—Jenna Self, MPH; Hawaii—Emily Roberson, MPH; Illinois—Theresa Sandidge, MA; Indiana–Jenny Durica, MPH; Iowa—Sarah Mauch, MPH; Louisiana—Jane Herwehe, MPH; Maine—Tom Patenaude, MPH; Maryland—Diana Cheng, MD; Massachusetts—Emily Lu, MPH; Michigan—Cristin Larder, MS; Minnesota—Judy Punyko, PhD, MPH; Mississippi—Brenda Hughes, MPPA; Missouri—Venkata Garikapaty, MSc, MS, PhD, MPH; Montana—JoAnn Dotson; Nebraska—Brenda Coufal; New Hampshire—Paulette Valliere, MPH; New Jersey—Lakota Kruse, MD; New Mexico—Eirian Coronado, MPH; New York State—Anne Radigan-Garcia; New York City—Candace Mulready-Ward, MPH; North Carolina—Kathleen Jones-Vessey, MS; North Dakota—Sandra Anseth; Ohio—Connie Geidenberger, PhD; Oklahoma—Alicia Lincoln, MSW, MSPH; Oregon—Kenneth

Rosenberg, MD, MPH; Pennsylvania—Tony Norwood; Rhode Island—Sam Viner-Brown, PhD; South Carolina—Mike Smith, MSPH; Texas—Rochelle Kingsley, MPH; Tennessee— Angela Miller, PhD, MSPH; Utah—Lynsey Gammon, MPH; Vermont—Peggy Brozicevic; Virginia—Marilyn Wenner; Washington—Linda Lohdefinck; West Virginia—Melissa Baker, MA; Wisconsin—Mireille Perzan, MPH; Wyoming—Amy Spieker, MPH.

## Author Contributions

**Conceptualization:** Rashida-E Ijdi, Janine Barden-O'Fallon.

**Data curation:** Rashida-E Ijdi.

**Formal analysis:** Rashida-E Ijdi.

**Methodology:** Rashida-E Ijdi, Janine Barden-O'Fallon.

**Supervision:** Janine Barden-O'Fallon.

**Validation:** Janine Barden-O'Fallon.

**Visualization:** Rashida-E Ijdi.

**Writing – original draft:** Rashida-E Ijdi.

**Writing – review & editing:** Rashida-E Ijdi, Janine Barden-O'Fallon.

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
