## [Decision Letter · Decision Letter 0]

21 Feb 2024

PONE-D-23-33931Association between intimate partner violence and postpartum contraceptive use in the United States– Evidence from PRAMS, 2016-2020PLOS ONE

Dear Dr. Ijdi,

Thank you for submitting your manuscript to PLOS ONE. After careful consideration, we feel that it has merit but does not fully meet PLOS ONE’s publication criteria as it currently stands. Therefore, we invite you to submit a revised version of the manuscript that addresses the points raised during the review process.

Thank you for this important submission that evaluates experiences of physical IPV from the PRAMS data. Both reviewers have provided extensive feedback that should be incorporated prior to resubmission. With your resubmission, please include a table that highlights each review, how the review was addressed, and the corresponding page number/line number. We look forward to receiving your revision.

We look forward to receiving your revised manuscript.

Kind regards,

Michelle L. Munro-Kramer, PhD, CNM, FNP-BC, FAAN

Academic Editor

PLOS ONE

Reviewers' comments:

Reviewer's Responses to Questions

**Comments to the Author**

1. Is the manuscript technically sound, and do the data support the conclusions?

Reviewer #1: Partly

Reviewer #2: Yes

2. Has the statistical analysis been performed appropriately and rigorously? 

Reviewer #1: I Don't Know

Reviewer #2: Yes

3. Have the authors made all data underlying the findings in their manuscript fully available?

Reviewer #1: No

Reviewer #2: Yes

4. Is the manuscript presented in an intelligible fashion and written in standard English?

Reviewer #1: Yes

Reviewer #2: Yes

5. Review Comments to the Author

Reviewer #1: 1. Suggest clarifying that this paper is about physical IPV in the title, as PRAMS also measures emotional and sexual IPV: “Association between intimate partner violence and postpartum contraceptive use in the United States– Evidence from PRAMS, 2016-2020”

Abstract

2. Please be very clear on what type of IPV is the focus of this paper and please be consistent throughout with terminology. There is switching from IPV to physical IPV. I suggest also using the term physical IPV.

3. PRAMS uses the terminology phases, rather than rounds. It should be PRAMS Phase 8.

4. Why was 2021 data not included? That was released in April 2023.

5. “Our sample included women who answered questions about the experience of physical IPV during and/or 12 months prior to their most recent pregnancy” – it’s not actually the most recent pregnancy. It’s the index birth from which the birth certificate was sampled. Someone could have had a pregnancy between the last birth and the timing of the survey.

6. “modern or natural method” how are you defining these categories? Why not just say use of any method?

7. This paper did not look at screening for physical IPV during pregnancy and/or its relationship to postpartum contraceptive use. The authors, however, suggest that “it is crucial to prioritize meeting unmet contraceptive needs as well as conducting universal screening for IPV among pregnant women.” Please align conclusions with the actual findings from this paper.

Introduction

8. “Research in the 1990s found that between 3% to 62 9% of pregnant women were estimated to have experienced IPV during pregnancy in the US, though the 63 real number could have been much higher than reported [13].” Why did the authors use this citation? These numbers were recently updated using PRAMS data…

a. Please see: D'Angelo DV, Bombard JM, Lee RD, Kortsmit K, Kapaya M, Fasula A. Prevalence of Experiencing Physical, Emotional, and Sexual Violence by a Current Intimate Partner during Pregnancy: Population-based Estimates from the Pregnancy Risk Assessment Monitoring System. J Fam Violence. 2022 Jan 14;38(1):117-126. doi: 10.1007/s10896-022-00356-y. PMID: 37205924; PMCID: PMC10193455.

9. “Mistimed or unwanted pregnancies can lead to induced abortion; an estimated 40% of unintended pregnancies end in abortion in the US [17]. Increases in correct and consistent contraceptive use, especially among populations at increased risk of having an unmet need for contraception, are therefore likely to lead to a decline in induced abortions.” Perhaps… but is this really a concern/goal? Induced abortion is not an adverse outcome and having an induced abortion could represent someone exerting their autonomy to do so and their reproductive preferences. Further, contraceptive use, alone, is not the solution to preventing unintended pregnancy. It is one tool that people can use to prevent pregnancy if that is their desire. I think this paper could use a good review of the framing to use more health equity/person centered language.

10. “However, to our knowledge, no recent studies have aimed at understanding the extent of the relationship of IPV on women’s contraceptive use in the US.” This is not true. The PRAMS team published an almost identical paper with data from Phase 7.

a. Stevenson AA, Bauman BL, Zapata LB, Ahluwalia IB, Tepper NK. Intimate Partner Violence around the Time of Pregnancy and Postpartum Contraceptive Use. Womens Health Issues. 2020 Mar-Apr;30(2):98-105. doi: 10.1016/j.whi.2019.11.006. Epub 2020 Jan 5. PMID: 31911042.

Methods

11. PRAMS is implemented in collaboration with state and local health departments

12. PRAMS isn’t always state level. NYC, DC and Puerto Rico are also included in Phase 8.

13. “These are the most recent available data containing information on IPV and contraceptive use behavior for women of reproductive age in the US” Why wasn’t 2021 data incorporated into this analysis? Then this sentence would be true.

14. Again, please use the terminology that PRAMS prefer – Phase 8

15. To clarify, PRAMS does not collect data on gender identity. Many PRAMS publications use the term “women”, so I think it’s okay. However, a limitation is that we don’t actually know the gender identity of people included in this analysis, only that they were listed as the birthing parent on the birth certificate. So it is possible that people who do not identify as cisgender women were included and they are being misgendered by the terms used in this paper.

16. It’s actually 2 – 6 months after delivery that people were mailed a survey.

17. The introduction and justification for the study focuses heavily on use of contraception as a means to avoid unintended pregnancy. However, the authors did not restrict the study sample to only people who wanted to avoid a next pregnancy. I suggest that the authors either do this, or add additional justification on why we would want people who want to achieve pregnancy use contraception. I think the authors are assuming this is the case, as the sample is people who gave birth in the prior 2-6 months, but many people in the US (30%) have interpregnancy intervals shorter than the recommended 18 months (although, 6 months is really the most adverse). I would suggest reframing in terms of why it is important women avoid short interpregnancy intervals.

18. PRAMS requires a minimum overall response rate threshold for the release of data for each year. Did all sites have data for all years included in this study? This should be specified. Some states only have data released for specific years, or even no years. So not all sites are included for every year available.

19. Why was “other” contraceptive methods recoded? There are responses that do actually fit within the other categories because of data collection errors. I suggest recoding as many as possible.

20. A huge limitation of using the method question is that lactational amenorrhea method (LAM) is not included. LAM does afford a good amount protection from pregnancy. You will find some responses like this in the “other” response category.

21. “This study does not have the scope to measure IPV beyond physical IPV as this information was not collected by the PRAMS core questions.” This is a study limitation; please move to discussion. Data is included on sexual and emotional IPV before, during, and after pregnancy, as well as physical IPV after pregnancy, but these are in the standard question set. You could have included this if you wanted to look at these associations.

22. What was the justification for grouping together AI/AN, Asian, etc in “other”? From Table 1, 25,242 people were included in this group. A best practice is to maintain a minimum set of categories (American Indian or Alaska Native, Asian, Black or African American, Native Hawaiian or Other Pacific Islander, and White), except when the sample size is so small that estimates would be unreliable.

23. How were people classified if they selected more than one race?

24. “The variable ‘state of residence’ was aggregated into four geographic 148 regions according to US Census divisions: Northeast, Midwest, South, and West [25].” Why? This likely absolves any association we would see between state of residence and access to contraception in the postpartum period.

25. Why wasn’t insurance status included as a covariate?

Results

26. “This contraceptive method-mix was calculated according to the efficacy of the last used method reported during the survey as described in the methods section.” This doesn’t need to be repeated.

27. Typo in this sentence: For the long-acting-reversible and permanent methods (LARC-PM) IUD and vasectomy use were hinger among IVP non-exposed group,”

28. “we found that women who had exposure to physical IPV in the last twelve months had 42% lower odds” This phrasing is incorrect. It’s people who experienced IPV in the 12 months prior to pregnancy and during pregnancy. It’s not the last 12 months.

29. “The association between IPV and contraceptive use remained the same for modern contraceptive use only which was tested separately (results not shown).” I think a better comparison would be methods that require prescription/interaction with a facility/provider/etc and those that don’t. A likely explanation is that people who experience violence may not seek healthcare from a provider (either due to fears of confidentiality, lack of trauma informed/high quality care, barriers to getting care, etc), and so they might use methods that don’t require a prescription or interaction with the healthcare system. HOWEVER, some of the methods that the authors classified as “natural” require partner support… periodic abstinence, withdrawal, etc. But people using these methods might not use consistently or might not be as exposed to sex as people not experiencing violence. I assume that breastfeeding and sexual frequency might also differed between the groups, and these are factors for pregnancy… and these were not controlled for.

30. Table 2: “Natural contraceptive methods are rhythm, withdrawal, abstinence, other etc.” Please clarify what is “etc”.

31. Authors should clarify all abbreviations in tables with a note (e.g., IUD)

32. “A main difference between women exposed to IPV and women who weren’t was in the use of birth control pills (14.4% vs. 7.7%, respectively) and condoms (18.8% vs. 9.0%, respectively).” But the authors did not test if these differences were statistically significant. Please rephrase or test each category.

Discussion

33. “Our study findings suggest that physical IPV during and around the time of pregnancy may impact postpartum contraceptive usage.” During and 12 months prior to pregnancy. The authors did not look at the postpartum period. Please clarify.

34. “Women who experienced physical IPV at the time of pregnancy” Please be consistent with terms.

35. “This is a significant issue, since IPV is associated with the risk of unintended pregnancy, sexually transmitted diseases, and other reproductive health concerns.” Yes, however, for these outcomes to occur you have to have exposure to sex. We see in Table 2 that ¼ of people who reported physical IPV are abstinent. Another 10% are sterilized/with a partner who is sterilized. It’s very difficult to know from this analysis if any of these people are actually at risk of unintended pregnancy and STDs.

36. “Comparable findings have been documented in earlier studies, such as in an analysis of PRAMS data from 2004 to 2008 found that women reporting IPV were significantly less likely to use postpartum contraception than women who did not (cOR, 0.66; 95% CI, 0.61-0.71) [31].”

a. Please see: Stevenson AA, Bauman BL, Zapata LB, Ahluwalia IB, Tepper NK. Intimate Partner Violence around the Time of Pregnancy and Postpartum Contraceptive Use. Womens Health Issues. 2020 Mar-Apr;30(2):98-105. doi: 10.1016/j.whi.2019.11.006. Epub 2020 Jan 5. PMID: 31911042.

b. This is the most up-to-date analysis with PRAMS data on this topic and should be incorporated.

37. “Our analyses indicated that women who experienced IPV had significantly lower usage rates for partner-dependent contraceptive methods such as condoms, natural family planning (including rhythm), 272 and vasectomy compared to their counterparts” . Where are these findings? The authors did not test if these different were statistically different from each other (Table 2).

38. “Our study found that women who have experienced physical IPV are more likely to rely on natural contraceptive methods-which is concerning as natural methods are less reliable and can be more difficult to use effectively, thus increasing exposure to unplanned pregnancy.” So in the previous sentence, the authors just said that there analysis indicated that IPV exposed women has less usage of partner dependent methods (rhythm, withdrawal, abstinence). Thus, this sentence contradicts that. This is also misleading because most of these women were using abstinence – meaning they do not have sex and thus cannot get pregnant. If you aren’t having sex, then you can’t be exposed to unplanned pregnancy.

a. Also, withdrawal is 78% effective. Fertility awareness methods range in effectiveness anywhere from low 70% to high 90%. Abstinence is 100% effective. The pill/condom/ring/etc. can also be difficult to use correctly, which is why, for example, typical effectiveness of condoms is only 87%. This is much higher than withdrawal, but withdrawal is still better than “nothing” and it’s great if it fits someone’s values and preferences at that point in time.

39. “The fact that nonuse of postpartum contraception is higher, and possibly increasing, among women who have experienced physical IPV...” The Stevenson et al. (2020) paper found that 21% of women exposed to IPV were not using contraception. This study found that 9% of women exposed to IPV were not using contraception. In terms of trends, it looks like postpartum contraceptive use has improved among women exposed to physical IPV.

40. Even among women not exposed to IPV, 94% is high. Last year, an analysis of Phase 8 PRAMS data was published on postpartum contraceptive use. They found that about 15% of people did not use contraception. So this study found an even smaller percentage of nonuse.

a. Bruce K, Stefanescu A, Romero L, Okoroh E, Cox S, Kieltyka L, Kroelinger C. Trends in Postpartum Contraceptive Use in 20 U.S. States and Jurisdictions: The Pregnancy Risk Assessment Monitoring System, 2015-2018. Womens Health Issues. 2023 Mar-Apr;33(2):133-141. doi: 10.1016/j.whi.2022.10.002. Epub 2022 Dec 1. PMID: 36464580.

41. “To address this issue, healthcare providers should adhere to recommended screening guidelines for IPV during pregnancy, and provide family planning, education, and resources to all patients, regardless of their disclosure.” Screening, alone, would likely not eliminate the effect of physical IPV on PPFP. However, preventing violence from occurring in the first place possibly could.

42. “In conclusion, our study underscores the importance of assessing IPV with the need of contraceptive methods as an important reproductive health outcome.” But this analysis didn’t consider whether women were actually screened and if that screening made a difference in contraceptive use. It doesn’t underscore the importance, it just shows that there is a relationship between IPV exposure and PPFP. There are many different interventions; screening could be one of them. Although, screening alone is not recommended by ACOG, USPSTF, or the CDC. These organizations recommend that IPV screening should always be coupled with referral and on-going support. It is likely that the referral and ongoing support make the difference, not screening alone.

43. “For example, those who participated in PRAMS were asked only a few general questions about IPV.” They asked exactly 2 questions.

44. “In addition, the term “abused” used in this survey, may be regarded as demeaning or judgmental”. The term “abused” is not used in the survey. The survey questions are:

a. “In the 12 months before you got pregnant with your new baby, did any of the following people push, hit, slap, kick, choke, or physically hurt you in any other way?”

b. “During your most recent pregnancy, did any of the following people push, hit, slap, kick, choke, or physically hurt you in any other way? “

45. Conclusion: again, screening alone will likely not impact reduced exposure to physical IPV or mediate this relationship with PPFP. There are a number of interventions that CDC, ACOG, and other organizations recommend to prevent violence before it occurs, especially for people who are pregnancy and postpartum.

a. Suggest reading: https://www.cdc.gov/violenceprevention/pdf/ipv-technicalpackages.pdf

Reviewer #2: The manuscript entitled, "Association between intimate partner violence and postpartum contraceptive use in the United States– Evidence from PRAMS, 2016-2020” is an important update on a topic that remains relevant. There are several aspects that commend this manuscript, including a generally clear writing style, and importance of topic. There are few minor typographical errors and some less than accurate phrasing. There are other considerations that should be examined prior to acceptance for publication. These are summarized below.

Lines 16-17 & 91: The PRAMS surveillance system is comprised of different “Phases” of the survey – not “rounds” as the authors state. As such, they utilized data from Phase 8 of the survey, not “round 8”.

Line 59: In the phrase “In the US 1 in 5… it should be a numeral 1 and not the letter I.

Lines 61-63: The authors cite research from the 90s. This discussion could have benefited from an examination of and comparison with more recent studies of a similar vein including the following.

1. Prevalence of Experiencing Physical, Emotional, and Sexual Violence by a Current Intimate Partner during Pregnancy: Population-based Estimates from the Pregnancy Risk Assessment Monitoring System.

D'Angelo DV, Bombard JM, Lee RD, Kortsmit K, Kapaya M, Fasula A.

J Fam Violence. 2022 Jan 14;38(1):117-126.

2. Intimate Partner Violence Before and During Pregnancy, and Prenatal Counseling Among Women with a Recent Live Birth, United States, 2009-2015.

Kapaya M, Boulet SL, Warner L, Harrison L, Fowler D.

J Womens Health (Larchmt). 2019 Nov;28(11):1476-1486.

Lines 76-77: The authors state that to their “knowledge, no recent studies have aimed at understanding the extent of the relationship of IPV on women’s contraceptive use in the US.” This is incorrect. There are a few fairly recent studies that have done just that including the reference below which is only cited in the discussion portion of this paper. The authors could have introduced Stevenson et al, 2020 here and used it to describe if or how their work differs and how it adds to prevailing literature.

"Intimate Partner Violence around the Time of Pregnancy and Postpartum Contraceptive Use.

Stevenson AA, Bauman BL, Zapata LB, Ahluwalia IB, Tepper NK.

Womens Health Issues. 2020 Mar-Apr;30(2):98-105.

Line 84: PRAMS is not an observational study. Please correct this.

Line 96: The word “survey” is missing after PRAMS.

Lines 96--97: This sentence is inaccurately stated. PRAMS was not “designed to collect a stratified sample of mothers who recently gave birth at participating sites using data from birth certificates” rather, PRAMS utilizes a stratified sample pulled from birth certificate records of women with a recent live birth in participating sites, to collect data on their experiences and behaviors before, during, and shortly after pregnancy.

Lines 124-125: The wording used here does not accurately reflect the PRAMS question cited which in fact reads "Are you or your husband or partner doing anything now to keep from getting pregnant? " If the authors choose to paraphrase the question, quotation marks should not be used as though citing directly.

Line 129: Please correct the typo in spelling of Mirena which is spelled “Minera”

Lines 134-135: The authors refence birth control methods here not previously listed in the methods they include in their outcome variable (lines 127-134), namely, tubectomy, vasectomy and abstinence.

Lines 138-140: The PRAMS survey has emotional and sexual abuse questions on the standard questions list. Why did the authors not request these standard questions? As reported in D'Angelo et al 2022, emotional IPV was the most frequently reported form of IPV, more prevalent than physical violence. It would have been beneficial to see if the patterns of contraceptive use differed by type of IPV.

Results

Line 178: Add the word ‘live’ before birth here (and anywhere else in the manuscript when describing the sample used.

Table 2, Lines 201-210: Again, the authors have included on the table and in the description of results, methods not referenced in the initial write up of the BC methods to be analyzed. They are reporting on tubectomy, vasectomy and abstinence which were never referenced in lines 127-134. Further, the authors introduce new terms in Table 2 “coil” and “loop” (types of IUD) not previously mentioned elsewhere. If these refer to the trade names of BC methods cited previously, they should either use coil and loop in parentheses against each respective trade name or remove the terms from the table and retain only the collective term “IUD” as used in their methods section.

Discussion

Line 264: Authors cite a “worsening trend” in view of results from data from 2004-2008. How do their results compare against more recently published similar studies such as Stevenson AA et al 2020?

Also, it would be interesting for the authors to give their thoughts around the higher prevalence of abstinence reported among those with IPV exposure compared to those without which is driving the higher prevalence of natural contraceptive methods in this group compared to modern methods. Particularly given it is a partner-dependent method.

General comment: Finally, I would suggest clarifying throughout the paper that the variable under question is PHYSICAL IPV. This is explained in the methods quite nicely, but the results, discussion and tables should clearly indicate that this study applies to physical IPV only.

6. PLOS authors have the option to publish the peer review history of their article (what does this mean?). If published, this will include your full peer review and any attached files.

Reviewer #1: No

Reviewer #2: No

---

## [Author Response · Author response to Decision Letter 0]

20 Apr 2024

April 19, 2024

RE: PONE-23-33931

Association between intimate partner violence and postpartum contraceptive use in the United States – Evidence from PRAMS 2016-2021

PLOS One

This manuscript follows PLOS One’s style requirements, including those for file naming according to the shared templates.

We have mentioned these issues in the ‘PRAMS Data’ and ‘Ethics approval’ section in this manuscript.

This study analyzed Pregnancy Risk Assessment and Monitoring Survey (PRAMS) data received from the U.S. Centers for Disease Control and Prevention (CDC). The data sources used in the analysis can be accessed at the weblink upon request: https://www.cdc.gov/prams/prams-data/researchers.htm. 

Our ethic statement provided in the method section of this study, page 8, lines 102 - 111: The PRAMS Phase 8 (2016-2021) data is released to the public and is provided for use by the approval of PRAMS sites and CDC over request [21]. Researchers requested de-identified PRAMS data by submitting a proposal to CDC with a data sharing agreement, and a standard application form. The study protocol was approved by the CDC and PRAMS participating state health departments by informed written consent to data request email on December 1, 2022, and the data was shared within four weeks after this approval. Ethical approval of this study was waived by the UNC Chapel Hill Office of Human Research Ethics Institutional Review Board IRB (Study# 22-2814) in a written consent on January 23, 2023, because the study was carried out using publicly available data that was anonymized and free of personally identifiable information. Only members of the research team listed on the application had access to the data. 

Reviewers' comments:

Review for PLOS One 

Reviewer #1: 

1. Suggest clarifying that this paper is about physical IPV in the title, as PRAMS also measures emotional and sexual IPV: “Association between intimate partner violence and postpartum contraceptive use in the United States– Evidence from PRAMS, 2016-2020”

We would like to thank the reviewer for this comment. In our study we focused on physical intimate partner violence (IPV) as this is covered by the core questionnaire. Therefor, in the revised manuscript we have updated the title as “Association between physical intimate partner violence and postpartum contraceptive use in the United States - Evidence from PRAMS 2016-2021”.

Abstract

2. Please be very clear on what type of IPV is the focus of this paper and please be consistent throughout with terminology. There is switching from IPV to physical IPV. I suggest also using the term physical IPV.

We would like to thank the reviewer for bringing out this point. We have corrected as required to be consistent with the terminology throughout the manuscript.

3. PRAMS uses the terminology phases, rather than rounds. It should be PRAMS Phase 8.

We would like to thank the reviewer for pointing out this. We have corrected the term as Phase 8 in this manuscript.

4. Why was 2021 data not included? That was released in April 2023. 

We requested CDC PRAMS data from 2016 to 2020, but were in actuality provided data from 2016 to 2021.Thus, in our study all the calculations were done with PRAMS data 2016 to 2021. We have corrected the text and included a table on State-wise data collection for each year between 2016 to 2021 (Annex 1).

5. “Our sample included women who answered questions about the experience of physical IPV during and/or 12 months prior to their most recent pregnancy” – it’s not actually the most recent pregnancy. It’s the index birth from which the birth certificate was sampled. Someone could have had a pregnancy between the last birth and the timing of the survey.

We have updated this sentence to read, “Inclusion criteria ….. 165,204 women.” , page 5, lines 121-127.

6. “modern or natural method” how are you defining these categories? Why not just say use of any method? 

We would like to thank the reviewer for your question. We have broadly categorized the methods of contraception as ‘modern method’ and ‘non-modern method’ (Hubacher & Trussel, 2015) to explain whether physical IPV has any impact in choosing postpartum contraceptive methods with the concept that women experiencing IPV may have different considerations and constraints when choosing between modern and non-modern methods. For instance, an abuser's control over a woman's autonomy may limit her access to healthcare services, thereby influencing her propensity to utilize natural methods over modern methods. 

Hubacher D, Trussell J. A definition of modern contraceptive methods. Contraception. 2015;92(5):420-421. doi:10.1016/j.contraception.2015.08.008

7. This paper did not look at screening for physical IPV during pregnancy and/or its relationship to postpartum contraceptive use. The authors, however, suggest that “it is crucial to prioritize meeting unmet contraceptive needs as well as conducting universal screening for IPV among pregnant women.” Please align conclusions with the actual findings from this paper. 

Thanks to the reviewer for pointing this out. We gave revised our suggesting as “It calls for the integration of IPV considerations into public health policies and clinical initiatives to improve maternal well-being.” In page 2, lines 33-34.

Introduction

8. “Research in the 1990s found that between 3% to 9% of pregnant women were estimated to have experienced IPV during pregnancy in the US, though the 63 real number could have been much higher than reported [13].” Why did the authors use this citation? These numbers were recently updated using PRAMS data…

a. Please see: D'Angelo DV, Bombard JM, Lee RD, Kortsmit K, Kapaya M, Fasula A. Prevalence of Experiencing Physical, Emotional, and Sexual Violence by a Current Intimate Partner during Pregnancy: Population-based Estimates from the Pregnancy Risk Assessment Monitoring System. J Fam Violence. 2022 Jan 14;38(1):117-126. doi: 10.1007/s10896-022-00356-y. PMID: 37205924; PMCID: PMC10193455.

We would like to thank the reviewer for providing us with this reference. We have updated this sentence with references in page 3, lines 55-58.

9. “Mistimed or unwanted pregnancies can lead to induced abortion; an estimated 40% of unintended pregnancies end in abortion in the US [17]. Increases in correct and consistent contraceptive use, especially among populations at increased risk of having an unmet need for contraception, are therefore likely to lead to a decline in induced abortions.” Perhaps… but is this really a concern/goal? Induced abortion is not an adverse outcome and having an induced abortion could represent someone exerting their autonomy to do so and their reproductive preferences. Further, contraceptive use, alone, is not the solution to preventing unintended pregnancy. It is one tool that people can use to prevent pregnancy if that is their desire. I think this paper could use a good review of the framing to use more health equity/person centered language.

Thanks to the reviewer for bringing up this point. We have deleted these two sentences, and added the following sentence “It is anticipated that the incidence of unintended pregnancies could be reduced through the proper and consistent use of contraceptives, especially within populations with a higher risk of experiencing unmet needs for contraception.” in page 3, lines 65-67. 

10. “However, to our knowledge, no recent studies have aimed at understanding the extent of the relationship of IPV on women’s contraceptive use in the US.” This is not true. The PRAMS team published an almost identical paper with data from Phase 7. 

a. Stevenson AA, Bauman BL, Zapata LB, Ahluwalia IB, Tepper NK. Intimate Partner Violence around the Time of Pregnancy and Postpartum Contraceptive Use. Womens Health Issues. 2020 Mar-Apr;30(2):98-105. doi: 10.1016/j.whi.2019.11.006. Epub 2020 Jan 5. PMID: 31911042.

Thank you for bringing this relevant publication to our attention. We appreciate your diligence in ensuring the accuracy of our statement regarding previous research on the relationship between intimate partner violence (IPV) and women’s contraceptive use in the United States. 

We acknowledge the existence of the article by Stevenson et al., 2020. This study indeed explores a similar topic to ours, examining the impact of IPV on postpartum contraceptive use using data from Phase 7 of the Pregnancy Risk Assessment Monitoring System (PRAMS) data between 2012 to 2015 in 36 states and 1 city. Our data is more recent with the wider coverage of 46 states and 4 cites. 

Methods

11. PRAMS is implemented in collaboration with state and local health departments

We would like to thank the reviewer for this comment. We have updated this information in page 4, line 84.

12. PRAMS isn’t always state level. NYC, DC and Puerto Rico are also included in Phase 8.

We have acknowledged this information in page 4, lines 86-87.

13. “These are the most recent available data containing information on IPV and contraceptive use behavior for women of reproductive age in the US” Why wasn’t 2021 data incorporated into this analysis? Then this sentence would be true. 

With reference to our clarification made to the reviewer’s comment above (#4), we now state that the data included the period of 2015-2021 of Phase 8.

14. Again, please use the terminology that PRAMS prefer – Phase 8

Thanks to the reviewer. We have updated the manuscript with the terminology ‘Phase 8’ throughout.

15. To clarify, PRAMS does not collect data on gender identity. Many PRAMS publications use the term “women”, so I think it’s okay. However, a limitation is that we don’t actually know the gender identity of people included in this analysis, only that they were listed as the birthing parent on the birth certificate. So it is possible that people who do not identify as cisgender women were included and they are being misgendered by the terms used in this paper. 

We appreciate the opportunity to address this important point. In response to this question, we included “Moreover, our analysis is based on individuals listed as the birthing parent on the birth certificates and may not fully encompass all gender identities.” as a limitation in our manuscript in page 16, lines 349-351. 

16. It’s actually 2 – 6 months after delivery that people were mailed a survey.

Many thanks to the reviewer for pointing this out. We have corrected. 

17. The introduction and justification for the study focuses heavily on use of contraception as a means to avoid unintended pregnancy. However, the authors did not restrict the study sample to only people who wanted to avoid a next pregnancy. I suggest that the authors either do this or add additional justification on why we would want people who want to achieve pregnancy use contraception. I think the authors are assuming this is the case, as the sample is people who gave birth in the prior 2-6 months, but many people in the US (30%) have interpregnancy intervals shorter than the recommended 18 months (although, 6 months is really the most adverse). I would suggest reframing in terms of why it is important women avoid short interpregnancy intervals. 

To address this comment from the reviewer we re-wrote as “Post-partum contraception plays ………prevalence of unintended pregnancies.” In pages 3-4, lines 65-72.

18. PRAMS requires a minimum overall response rate threshold for the release of data for each year. Did all sites have data for all years included in this study? This should be specified. Some states only have data released for specific years, or even no years. So not all sites are included for every year available.

Several PRAMS sites have data released for specific years or may have no data available for certain years within our study period. In our revised manuscript, we have included a section in the methodology detailing the availability of data from each PRAMS site for the years included in our study. 

In the manuscript, we included “A comprehensive list of available datasets for Phase 8 (year 2016-2021), categorized by city and state, is provided as an annex alongside this manuscript. The threshold for the minimum overall response rate was set at 55% for 2016 -2017, and 50% for 2018-2021.” in lines 105-108, pages 4-5 according to the reference (https://www.cdc.gov/prams/prams-data/researchers.htm).

19. Why was “other” contraceptive methods recoded? There are responses that do actually fit within the other categories because of data collection errors. I suggest recoding as many as possible.

The "other" contraceptive methods were recoded to ensure clarity and accuracy in analyzing the data.. From the category ‘other’ we have recoded as many answers as possible to the other categories of contraception, which did not make any changes to estimates due to the low number. 

20. A huge limitation of using the method question is that lactational amenorrhea method (LAM) is not included. LAM does afford a good amount of protection from pregnancy. You will find some responses like this in the “other” response category. 

Thank you for your valuable feedback regarding the omission of the lactational amenorrhea method (LAM) from the listed contraceptive methods in our study. The decision to not explicitly categorize LAM as a separate contraceptive method in our analysis was based on the constraints of the PRAMS questionnaire design and the categorization of responses. As you correctly pointed out, responses that might pertain to LAM use were likely captured under the "other" response category, which, unfortunately, does not allow for a detailed analysis of LAM's specific prevalence and impact.

In light of your comment, we have updated our manuscript adding in the limitation “LAM was not included …………….accessibility” On page 16, lines 358 – 363.

21. “This study does not have the scope to measure IPV beyond physical IPV as this information was not collected by the PRAMS core questions.”

---

## [Decision Letter · Decision Letter 1]

16 Jun 2024

PONE-D-23-33931R1Association between intimate partner violence and postpartum contraceptive use in the United States– Evidence from PRAMS 2016-2020PLOS ONE

Dear Dr. Ijdi,

Thank you for submitting your manuscript to PLOS ONE. After careful consideration, we feel that it has merit but does not fully meet PLOS ONE’s publication criteria as it currently stands. Therefore, we invite you to submit a revised version of the manuscript that addresses the points raised during the review process.

The Revised manuscript has been re-reviewed by two reviewers and their comments are available below and in the attached document. Reviewer 1 has raised concerns that some of the points in your rebuttal from the previous round of review have not been included in the revised manuscript main text. They have also provided some recommendations around framing of the manuscript to improve its quality.

We look forward to receiving your revised manuscript.

Kind regards,

Emma Campbell, Ph.D

Staff Editor

PLOS ONE

Journal Requirements:

Reviewers' comments:

Reviewer's Responses to Questions

**Comments to the Author**

1. If the authors have adequately addressed your comments raised in a previous round of review and you feel that this manuscript is now acceptable for publication, you may indicate that here to bypass the “Comments to the Author” section, enter your conflict of interest statement in the “Confidential to Editor” section, and submit your "Accept" recommendation.

Reviewer #1: (No Response)

Reviewer #2: All comments have been addressed

2. Is the manuscript technically sound, and do the data support the conclusions?

Reviewer #1: (No Response)

Reviewer #2: Yes

3. Has the statistical analysis been performed appropriately and rigorously? 

Reviewer #1: (No Response)

Reviewer #2: Yes

4. Have the authors made all data underlying the findings in their manuscript fully available?

Reviewer #1: (No Response)

Reviewer #2: Yes

5. Is the manuscript presented in an intelligible fashion and written in standard English?

Reviewer #1: (No Response)

Reviewer #2: Yes

6. Review Comments to the Author

Reviewer #1: (No Response)

Reviewer #2: I have no additional comments to make. The authors have provided responses to all observations made in the original manuscript.

7. PLOS authors have the option to publish the peer review history of their article (what does this mean?). If published, this will include your full peer review and any attached files.

Reviewer #1: No

Reviewer #2: No

---

## [Author Response · Author response to Decision Letter 1]

16 Sep 2024

September 11, 2024

RE: PONE-23-33931

Association between physical intimate partner violence and postpartum contraceptive use in the United States – Evidence from PRAMS 2016-2021

PLOS One

Minor Revisions Required

Review comments 

General comment: Thank you for responding to my previous comments. I still have concerns after reading the revision. I am flagging for the editor that some revisions were not made in the manuscript text. I am also noting some issues with the framing of this paper. Please see my notes below. I believe this paper could benefit from a thorough review of the US literature on the relationship between IPV and contraceptive use (with a focus in the postpartum period). I also believe this paper could be reframed using a reproductive justice lens, rather than some of the instrumental arguments it seems to be using as justification.

This paper is actually opportune to call attention to the ways in which violence, specifically physical IPV, plays a role in people’s ability to have children, not have children, and raise children in safe and sustainable environments. It draws on almost no perspectives from the violence prevention literature as well, which I see as a huge flaw. If IPV is associated with PPFP, then the conclusions should reflect that and call to invest in programming to reduce and prevent IPV before it occurs, and perhaps offer some evidence-based interventions or way to intervene and prevent IPV (so that postpartum people can make decisions with full autonomy). 

Specific comments:

1. We have broadly categorized the methods of contraception as ‘modern method’ and ‘non-modern method’ (Hubacher & Trussel, 2015) to explain whether physical IPV has any impact in choosing postpartum contraceptive methods with the concept that women experiencing IPV may have different considerations and constraints when choosing between modern and non-modern methods. For instance, an abuser's control over a woman's autonomy may limit her access to healthcare services, thereby influencing her propensity to utilize natural methods over modern methods. Hubacher D, Trussell J. A definition of modern contraceptive methods. Contraception. 2015;92(5):420-421. doi:10.1016/j.contraception.2015.08.008

a. Hubacher & Trussell (2015) is more applicable to the international context. Additionally, their classification of lactational amenorrhea method (LAM) as “non modern” is outdated and conflicts with newer classification systems that recognize LAM as a highly effective method for postpartum people. The same is true for standard days method (SDM) which is also considered a modern method in the context of LMICs (and I believe in the US). I suggest the authors draw on US literature, considering the sample is US postpartum people. 

b. The above rational that the authors provide does not consider that “nonmodern” methods like withdrawal require a tremendous amount of support from male partners. Likewise, past research suggests that fertility awareness-based methods (FABs) also require partner support, because often they require the partner to agree to periods of monthly abstinence. Some modern methods can also be used covertly. There is an entire body of literature on covert use by women experiencing IPV. I suggest the authors reframe to the US context, and again suggest the authors consider any method of family planning considering that the goal of FP programs and services is to support people in using whatever method matches their values and preferences, not just modern methods.

c. I am also noting that the authors seemed to upload a version of their response to the reviews with comments. I see in the comments the authors noting that: “Modern” family planning (FP) methods refer to the following: pill, intrauterine device, implant, injectable, condom (male and female), spermicide, diaphragm, patch, vaginal ring, sponge, cervical cap, lactational amenorrhea method, emergency contraception, standard days method, basal body temperature method, TwoDay method, sympto-thermal method, and sterilization (tubal ligation and vasectomy).” Why wasn’t this definition of modern methods used? I believe it is much more consistent with the US literature and classification systems. 

d. I am also flagging that there is a focus on use on modern contraception… but then the results seem to be about any method of contraception? This is very confusing for the reader. I would suggest being clear about what is the focus of this paper.

We would like to thank the reviewer for the detailed discussions on the categorization of contraceptive methods. After reviewing the results, we have decided to drop the analysis comparing use of modern with non-modern, and keep only ‘any method’ to continue with the paper. 

Therefore, we have deleted “However, women exposed to physical………. Opposite trend (56% and 69%, respectively).” From abstract from page 1, lines 24-26.

Deleted “These contraceptive methods ……… for the analysis.” In page 6, lines 149 to 156 form the method section.

Additionally deleted “The use of modern ….. level (p<0.001).” in page 10, lines 250 to 255, and “The association …… tested separately.” In page 11, lines 288 to 289 from the result section. 

We also have deleted ‘categories of contraceptive methods’ from Table 2.

2. “Thanks to the reviewer for bringing up this point. We have deleted these two sentences, and added the following sentence “It is anticipated that the incidence of unintended pregnancies could be reduced through the proper and consistent use of contraceptives, especially within populations with a higher risk of experiencing unmet needs for contraception.” in page 3, lines 65-67”

a. I am flagging that I did not find this sentence in the manuscript. I am flagging for the editor to make sure that all changes that the authors reported were in fact reflected in the paper.

We would like to thank the reviewer for flagging the absence of this sentence. We apologize that this sentence was revised in the manuscript, but not updated in the rebuttal. The updated sentence was “It is anticipated that the effective and consistent use of contraceptives has the potential to decrease the unmet need for postpartum contraception, particularly among populations with higher prevalence of unintended pregnancies [18].” In page 4, lines 69-72. 

b. This new sentence needs a citation. Please also define what populations are at higher risk of experiencing unmet need for contraception in the US context. 

We have included reference for this revised statement with the following citation,:

Schummers L, Hutcheon JA, Hernandez-Diaz S, et al. Association of Short Interpregnancy Interval With Pregnancy Outcomes According to Maternal Age. JAMA Intern Med. 2018;178(12):1661–1670.

Here, we have defined the population as ‘who had higher prevalence of unintended pregnancies’.

3. “The rationale is to provide information for policy makers and stakeholders in planning and implementing policies to boost utilization of postpartum contraceptives to improve maternal health with specific focus on the physical IPV exposed women.” 

a. How does use of contraception improve maternal health in the US context? This is not clear from this sentence of the preceding text about unintended pregnancy and nonuse of contraception. In the US, recent studies have shown that the evidence is even mixed on the casual effect of short interpregnancy intervals on some maternal health outcomes. ACOG provides a good overview here: https://www.acog.org/clinical/clinical-guidance/obstetric-care-consensus/articles/2019/01/interpregnancy-care#:~:text=Because%20the%20interpregnancy%20period%20is,a%20continuum%20from%20postpartum%20care.

We would like to thank to the reviewer for this comment. We have revised the statement as “The rationale of this study is to inform policymakers and stakeholders in planning and implementing policies aimed at improving reproductive and maternal health, with a specific focus on enhancing postpartum contraceptive use among women exposed to physical IPV in the United States.” in page 4, lines 80 to 83.

b. Why would or should the goal of PPFP programs and services be to increase uptake of postpartum contraception? If the goal is to improve maternal health and/or reduce violence (as the authors state), there are numerous, effective interventions that can improve the availability and quality of maternal and infant health services and a number of strategies that can be implemented proven to prevent IPV in the US. 

As noted above, the text referring to increasing uptake of PPFP was removed.

4. “Post-partum contraception plays a crucial role in safeguarding women’s health by effectively preventing short interpregnancy intervals… minimizing associated health complications [9,15].” 

a. This sentence seems to have been added in the update. References 9 and 15 are about the relationship between violence and reproductive health broadly. If the authors want this sentence included, they should provide a citation to support claims that postpartum contraception prevents short interpregnancy intervals in the US context. 

We would like to thank the reviewer for this comment. We included this sentence in the previous revised version. Here we have updated the citation.

5. “Moreover, our analysis is based on individuals listed as the birthing parent on the birth certificates and may not fully encompass all gender identities.””

a. I appreciate that the authors added this as a limitation, but I think it’s missing the issue. The analysis cannot establish gender identity because the survey does not ask about gender identity. Thus, it cannot be assumed that it is only cis-gender women in this analysis. It could be that there are some transmen or other individuals of diverse genders included in this study, and so use of the term “woman/women/mothers” may not accurately represent the study population. In addition, findings and conclusions cannot be generalized to LGBTQ+ populations. It is also difficult to disentangle and account for infertility in this population. Some respondents state that they used ART to become pregnant (either because of infertility or because of same sex relationships) in open-ended questions. However, the analysis doesn’t account for that and so there may be some people who are not using family planning because they are not at risk of pregnancy. 

We would like to thank the reviewer for this detailed comment. We used the term ‘woman/women/mother’ in our manuscript to represent the sample population of this survey who had given birth between 2 and 6 months before the survey was conducted. The PRAMS questionnaire also uses term ‘women’ in their questionnaire/topic reference document (reference is given below).

PRAMS Phase 8 Topic Reference Document. Weblink: https://www.cdc.gov/prams/pdf/questionnaire/Phase-8-Topics-Reference_508tagged.pdf

Moreover, previous articles using PRAMS data have also used the term ‘women’ explaining for their sample population in their manuscripts (Kapaya et al., (2019); Alhusen et al., (2015)).

Alhusen, J. L., Ray, E., Sharps, P., & Bullock, L. (2015). Intimate partner violence during pregnancy: maternal and neonatal outcomes. Journal of women's health (2002), 24(1), 100–106. https://doi.org/10.1089/jwh.2014.4872

Kapaya, M., Boulet, S. L., Warner, L., Harrison, L., & Fowler, D. (2019). Intimate Partner Violence Before and During Pregnancy, and Prenatal Counseling Among Women with a Recent Live Birth, United States, 2009-2015. Journal of women's health (2002), 28(11), 1476–1486. https://doi.org/10.1089/jwh.2018.7545

b. “The PRAMS survey employes a stratified sample extracted from birth certificate records of women who have recently given birth in participating sites. This sample is used to collect data on their experiences and behaviors before, during, and shortly after pregnancy. Women were mailed a questionnaire within 2–6 months following delivery, and those who did not respond to repeated mailings were contacted by 105 telephone. Participants were consented prior to completing the PRAMS survey per CDC protocols [21].”

i. Here is an example pulled from the methods. The authors state that the sample is drawn from birth certificate records of women who recently have a given birth. Gender identity is not asked in birth certificate records. It is the person who was recorded as the infant’s mother that was sampled. We actually don’t know how this person identifies, so by the authors calling them “women” it could be misgendering people. Similarly, it is people who gave birth, not necessarily “women”, who were mailed the questionnaire. 

To address this comment, we have included “Moreover, our analysis is based on individuals listed as the birthing parent on the birth certificates and may not fully encompass all gender identities.” as a limitation in our manuscript in page 16, lines 349-351 as a limitation in our manuscript.

ii. Also, it’s a sample of live births.

Thanks to the author for pointing this out. We have corrected it.

6. I appreciate the rationale for race/ethnicity groups that was provided in the response. However, it is best practice to provide the sample size for each group. Please provide rationale for the race/ethnicity categories (Black non-Hispanic, White non-Hispanic, all other races non-Hispanic, and Hispanic) in the manuscript text with sample sizes for all the race categories that were collapsed into other. 

We would like to thank the reviewer for this comment. We have updated race/ethnicity as “race/ethnicity (categorized as ‘Black non-Hispanic’, ‘White non-Hispanic’, all other races (e.g., Other Asian (4.3%), American Indian (3.6%), Chinese (1.2%), Japanese (0.2%), Philipino (0.8%), Hawaiian (0.1%), Ak Native (0.8%), mixed race (5.7%), and other non-White (5.5%))” in page 7, lines 175-176. In the result section we have added the all the race categories that were collapsed into ‘other’ in page 9, lines 229 to 231.

7. I appreciate the rationale for the groups of states. I advise the authors to review the PRAMS methodology, as well as PRAMS publications. It is common for PRAMS analysis to include each individual state. Also, it did not come through in reading the paper that the authors wanted to make claims about geographic differences. “this geographic representation allows for easier interpretation and comparison of our findings with existing literature and public health data”. Please add these interpretations to the results section. This would greatly benefit the paper if that was an objective of the analysis. Please also add this justification to the methods section.

We would like to thank the review for these comments. In the methodology section we also included “with the aim to reduce the potential confounding effects of individual state-level differences that many not be pertinent to the scope of this study.” After categorizing state of residences in page 7, lines 174-175. Findings on regional variation are presented for Figure 2.

We have added “Moreover, the prevalence of physical IPV varies significantly across different states in the USA. Some states have more robust support systems and legal protections for IPV survivors, which can influence reporting rates and overall prevalence of IPV [41].” In the discussion section, in page 16, lines 351 to 353.

8. The rationale for exclusion of insurance status is not clear. Did the authors test insurance status for multicollinearity with other covariates? I don’t see this in the methods or in the response to reviewers. 

The decision not to include insurance status as a covariate was based on the direct acyclic graph (DAG) used in the study design. The DAG identified a minimally sufficient adjustment set of covariates that were considered essential for controlling confounding in the relationship between intimate partner violence (IPV) and postpartum contraceptive use. The focus of this study is on demographic, socioeconomic, and reproductive health variables that were theoretically and empirically supported as significant confounders based on prior literature. Insurance status was not identified as a necessary covariate within this context and was therefore not included in the final model. This decision also aligns with the aim of creating a simplified model to avoid excessi

---

## [Editor Report · Decision Letter 2]

1 Nov 2024

PONE-D-23-33931R2Association between physical intimate partner violence and postpartum contraceptive use in the United States – Evidence from PRAMS 2016-2021PLOS ONE

Dear Dr. Rashida-E Ijdi,

Thank you for submitting your manuscript to PLOS ONE. After careful consideration, we feel that it has merit but does not fully meet PLOS ONE’s publication criteria as it currently stands. Therefore, we invite you to submit a revised version of the manuscript that addresses the points raised during the review process.

**ACADEMIC EDITOR: **Minor amendments Required

Thanks for conducting such a good research. I am concerned that under the conclusions more detailed recommendations are needed with greater specificity. Since IPV is the current burning question and the study affects reproductive rights of women, it requires more recommendations to solve the problem. I would be interested to know whether you are linked to the international human right to reproductive rights. Otherwise, this manuscript written in good English and adds value to the existing science.

Please share with us the ethical approval you have used to conduct this research if it has not been done before.

We look forward to receiving your revised manuscript.

Kind regards,

Kahsay Zenebe Gebreslasie, MSc

Academic Editor

PLOS ONE

Journal Requirements:

Additional Editor Comments:

Minor amendments Required

Thanks for conducting such a good research. I am concerned that under the conclusions more detailed recommendations are needed with greater specificity. Since IPV is the current burning question and the study affects reproductive rights of women, it requires more recommendations to solve the problem. I would be interested to know whether you are linked to the international human right to reproductive rights. Otherwise, this manuscript written in good English and adds value to the existing science.

Please share with us the ethical approval you have used to conduct this research if it has not been done before.

---

## [Author Response · Author response to Decision Letter 2]

14 Nov 2024

Minor amendments Required

Thanks for conducting such a good research. I am concerned that under the conclusions more detailed recommendations are needed with greater specificity. Since IPV is the current burning question and the study affects reproductive rights of women, it requires more recommendations to solve the problem. I would be interested to know whether you are linked to the international human right to reproductive rights. Otherwise, this manuscript written in good English and adds value to the existing science.

Thank you for your thoughtful feedback and for recognizing the value of our research. We truly appreciate your suggestions and agree that providing more detailed and specific recommendations in the conclusions section is crucial, especially given the importance of addressing IPV and its impact on reproductive rights. 

The updated conclusion of the paper is written as “The study findings reveal the impact of physical IPV on postpartum contraceptive use, highlighting the need for IPV screening during pregnancy including the provision of person-centered post-partum family planning education and counseling, which is vital for ensuring women’s reproductive autonomy and well-being. These findings also call for integrating IPV prevention into maternal health strategies and enhancing support for affected women, building on existing policies. Further research should continue to explore the long-term effects of IPV on maternal health to inform clinical and policy initiatives further.” As echoing the detailed recommendation we mentioned in the discussion section of this manuscript.

---

## [Editor Report · Decision Letter 3]

19 Nov 2024

Association between physical intimate partner violence and postpartum contraceptive use in the United States – Evidence from PRAMS 2016-2021

PONE-D-23-33931R3

Dear Dr. Rashida-E Ijdi,

We’re pleased to inform you that your manuscript has been judged scientifically suitable for publication and will be formally accepted for publication once it meets all outstanding technical requirements.

Kind regards,

Kahsay Zenebe Gebreslasie, MSc

Academic Editor

PLOS ONE
---

## [Editor Report · Acceptance letter]

1 Dec 2024

PONE-D-23-33931R3 

PLOS ONE

Dear Dr. Ijdi, 

I'm pleased to inform you that your manuscript has been deemed suitable for publication in PLOS ONE. Congratulations! Your manuscript is now being handed over to our production team.

Kind regards, 

on behalf of

Mr. Kahsay Zenebe Gebreslasie 

Academic Editor

PLOS ONE